# Improved YOLO-V3 with DenseNet for Multi-Scale Remote Sensing Target Detection

**DOI:** 10.3390/s20154276

**Published:** 2020-07-31

**Authors:** Danqing Xu, Yiquan Wu

**Affiliations:** College of Electronic and Information Engineering, Nanjing University of Aeronautics and Astronautics, Nanjing 211106, China; xudanqing@163.com

**Keywords:** remote sensing image, target detection, multi-scale, YOLO-V3, convolutional neural network, DenseNet

## Abstract

Remote sensing targets have different dimensions, and they have the characteristics of dense distribution and a complex background. This makes remote sensing target detection difficult. With the aim at detecting remote sensing targets at different scales, a new You Only Look Once (YOLO)-V3-based model was proposed. YOLO-V3 is a new version of YOLO. Aiming at the defect of poor performance of YOLO-V3 in detecting remote sensing targets, we adopted DenseNet (Densely Connected Network) to enhance feature extraction capability. Moreover, the detection scales were increased to four based on the original YOLO-V3. The experiment on RSOD (Remote Sensing Object Detection) dataset and UCS-AOD (Dataset of Object Detection in Aerial Images) dataset showed that our approach performed better than Faster-RCNN, SSD (Single Shot Multibox Detector), YOLO-V3, and YOLO-V3 tiny in terms of accuracy. Compared with original YOLO-V3, the mAP (mean Average Precision) of our approach increased from 77.10% to 88.73% in the RSOD dataset. In particular, the mAP of detecting targets like aircrafts, which are mainly made up of small targets increased by 12.12%. In addition, the detection speed was not significantly reduced. Generally speaking, our approach achieved higher accuracy and gave considerations to real-time performance simultaneously for remote sensing target detection.

## 1. Introduction

Recently, remote sensing images [1,2,3,4] have attracted more research in the field of computer version (CV) with the rapid development of satellite and imaging technology. There is a significant value on information extraction of remote sensing images. Remote sensing target detection [5,6,7] has important and extensive applications in military, navigation, salvage, and other aspects, which requires high speed and accuracy for target detection algorithms.

The rapid development of computer technology makes it possible for the applications of the convolutional neural network (CNN) [8,9,10,11], which requires high computing power. Compared with traditional target detection algorithms like HOG-SVM (Histogram of Oriented Gradients-Support Vector Machine) [12,13], DPM (Deformable Parts Model) [14,15], and HOG-Cascade [16,17], CNN-based target detection algorithms have great advantages in many aspects such as speed and accuracy. Convolutional neural network (CNN) is a kind of feed forward neural network with convolutional computing and it usually has a deep structure. It is one of the most important components of deep learning [18,19,20]. Recently, the research of deep learning in target detection has become a hot spot. The CNN-based target detection models can mainly be divided into two categories, which include the two-stage ones and the one-stage ones.

Currently, the two-stage ones are represented by R-CNN [21], and then Fast R-CNN [22,23], Faster R-CNN [24,25], and Mask R-CNN [26,27], which have been developed on the basis of it. As the name implies, the two-stage target detection algorithms divide the detection processes into two steps. First, the Region Proposed Network (RPN) [28,29,30] is used to extract the information of the targets and then the detection layers predict location and category information of the targets. The other ones are one-stage target detection algorithms including SSD (Single Shot Multibox Detector) [31,32,33], DSSD (Deconvolution Single Shot Multibox Detector) [34], FSSD (Feature Fusion Single Shot Multibox Detector) [35], YOLO [36], YOLO-V2 [37], and YOLO-V3 [38]. Instead of using the region proposed network (RPN), the one-stage algorithms obtain the predictive information of location and category directly. Therefore, they are also called the regression-based algorithms and they can usually achieve higher detection speed than the two-stage ones. At present, numerous state-of-the-art target detection models with higher speed are proposed based on YOLO such as YOLO-V3 tiny [39] and TF-YOLO [40]. Therefore, the accuracy of them is not satisfactory.

From the current research, remote sensing target detection usually faces the following challenges: one is that remote sensing targets are usually small and take up fewer pixels, which makes it difficult to extract features. The second challenge is that remote sensing images are usually disturbed by shadow, light, and other external factors. In addition, the scales of the remote sensing targets are usually different. To solve these problems, researchers have made unremitting efforts.

In order to realize remote sensing target detection, the inchoate research is mainly based on template matching, which is to match the target with a specific template for detection. For example, Weber et al. [41] proposed a method of making use of image analysis to extract coastline templates and adopted this method to detect oil tanks. It achieved good results. However, although the method of template matching is simple and effective, its overall robustness is poor and it is sensitive to the shapes of the targets and geometric deformation. The algorithms based on image analysis are to judge whether each region of the remote sensing image has a target by segmentation and classification. For example, Feng et al. [42] proposed the algorithm named multi-resolution segmentation, which segmented remote sensing images into multiple regions for detection by three parameters: shape, scale, and density. Compared with the method of template matching, this method is more flexible and can combine contextual semantic information, which has achieved good results in some tasks. However, this kind of algorithm still needs to be designed manually for segmentation, which is not universal.

Compared with the previous two algorithms, the remote sensing target detection algorithms based on deep learning have better accuracy and robustness because they no longer use the features of manual design. Sun et al. [43] extracted the region of interest with sliding windows, and then used the features of Bag-of-Words to detect the targets. Zhang et al. [44] and Yu et al. [45] combined the prior characteristics of the airports and coasts, respectively, with deep learning to conduct remote sensing target detection. More commonly, researchers use existing target detection algorithms such as Faster R-CNN in remote sensing target detection tasks. However, when these models are applied to remote sensing target detection tasks, their performance is poor due to the factors such as illumination, cloud cover, and complex background interference.

As an advanced target detection model, YOLO-V3 adopts a feature pyramid network (FPN) [46,47], ResNet (Residual Network) [48], and achieves good performance in speed and accuracy. YOLO-V3 predicts targets at three different scales. Compared with the previous two versions, YOLO-V3 enhances the capability of detecting multi-scale targets, especially small targets. Abundant improved algorithms have been proposed since YOLO-V3 came out. References [49,50] adopted four detection layers to enhance the performance of detecting small targets. Reference [51] adopted circular ground truth to realize tomato detection. Reference [52] increased another shortcut connection to concatenate 2 CBLs (Convolution-Batch Normalization-Leak ReLU) between two ‘residual units’ to enhance the performance for the feature extraction network of information transfer. Reference [40] simplified the feature extraction network to obtain faster detection speed and adopted multiple layers concatenation to enhance the performance of feature extraction. The above improved algorithms achieved a good detection effect. However, the resolution of remote sensing images is large. The scales of remote sensing targets are small and the backgrounds are complex. These algorithms, which have excellent performance on routine datasets, are not suitable for remote sensing target detection. Therefore, we need to design a feature extraction network and detection networks for our proposed algorithm elaborately.

According to the characteristics of remote sensing targets, the proposed method was improved based on the YOLO-V3 model. The main contributions in this paper include the following. (1) In order to reduce reliance on ResNet and enhance the ability of feature information extraction, which is inspired by DenseNet, improved densely connected units proposed to replace some of the residual units of Darknet53. (2) To further improve the ability of detecting multi-scale remote sensing targets, we extended the original three output layers of YOLO-V3 to 4. (3) In order to avoid gradient vanishing, instead of five convolutional layers in each detection layers, three residual units were adopted. The experimental results on remote sensing images show that the proposed method not only has good performance in accuracy, but also gives attention to real-time performance for remote sensing target detection.

The rest of this paper is as follows. In Section 2, we introduced the theory of YOLO and the framework of YOLO-V3. In Section 3, we described the improved method of our approach in details. Section 4 gives the experiments of the proposed algorithm on the RSOD dataset and compared the performance of our approach with other classical algorithms. Lastly, the conclusion is shown in Section 5.

## 2. The Theory of YOLO

YOLO (You Only Look Once) is a kind of one-stage algorithm, which transforms target detection as a regression problem. Compared with Faster R-CNN, YOLO obtain the predictive information of location and categories directly without a region proposed network (RPN). After continuous development, YOLO has been developed from YOLO-V1 to YOLO-V2 and the latest YOLO-V3.

### 2.1. The Principle of YOLO

At the beginning, the network divides each input image into S×S grid cells. The grid, which center on the ground truth (GT) of the target falls in, is responsible for detecting it. Each grid cell defines *B* bounding boxes as well as their corresponding confidence scores. Each bounding box contains *C* classes. We denote them as $P\left(Class_{i}|Object\right)$. If the center of the target falls in the grid cell, then P(Object)=1. Otherwise, P(Object)=0. The confidence score is defined as: $P\left(Object\right)\times {\mathrm{IOU}}_{pred}^{truth}$. It reflects the probability that the grid cell contains targets and the accuracy that the bounding box predicts. IOU represents the overlap area between the bounding box and the ground truth (GT). The class-specific scores can be denoted in Equation (1).
(1)$$P\left(Class_{i}|Object\right)\times P\left(Object\right)\times {\mathrm{IOU}}_{pred}^{truth}=P\left(Class\right)\times {\mathrm{IOU}}_{pred}^{truth}$$

YOLO has made greater achievements than Faster R-CNN in terms of speed, but it also brings the low accuracy of detection. On the basis of YOLO-V1, YOLO-V2 introduces the concept of the anchor box and runs k-means on the dataset to generate appropriate prediction boxes at the beginning. Instead of full connected layers (FC), YOLO-V2 introduces convolutional layers in the output end. In addition, YOLO-V2 also adopts Batch Normalized, New feature extraction network (Darknet19), which greatly improves the performance compared with YOLO-V1.

YOLO-V3 is a further improved version based on YOLO-V2 by upgrading the original Darknet19 to Darknet53 and adopts multi-scale detection layers (three scales) to detect the targets. This allows YOLO-V3 to detect small targets more effectively.

### 2.2. The Network of YOLO-V3

YOLO-V3 adopts Darknet53 as its feature extraction network. In order to prevent information loss caused by pooling layers, Darknet53 adopts a full convolutional network (FCN). The network is basically made up of convolutional kernels of 1 × 1 or 3 × 3. Since it contains 53 convolutional layers, it is called Darknet53. In order to extract deeper features and avoid gradient fading by drawing on the residual network, Darknet53 added five residual modules to the network in which each was composed of one or multiple residual units.

YOLO-V3 borrows the idea of the feature pyramid network (FPN). The network carries out five times of the down-sampling processing on each input image. The output feature map of the feature extraction is down-sampled by 32×, which means the output feature map is 1/32 of the size of the input image. Then YOLO-V3 transmits the last three down-sampled layers to the detection layers for target detection. The network of YOLO-V3 predicts at three scales. The sizes of the three scales are 13 × 13, 26 × 26, and 52 × 52, which are responsible to detection big targets, medium-sized targets, and small targets, respectively. The deep-level feature maps contain a mass of semantic information while the shallow-level feature maps contain a mass of fine-grained information. Therefore, to carry out feature fusion, the network uses up-sampling to keep the size of the feature map down-sampled by 32×, which is consistent with the feature map down-sampled by 16×, and then merges the feature maps by concatenation. Similarly, we do the same for the feature map down-sampled by 16× and the feature map down-sampled by 8×. The structure of YOLO-V3 and its feature extraction network are shown in Figure 1 and Table 1, respectively.

## 3. Related Work

### 3.1. Improved Densely Connected Network

The improvement of You Only Look Once (YOLO)-V3 is mainly based on the concept of a residual network. Darknet53 uses several residual units, and the ResNet made up of these residual units contains a large number of parameters and it is responsible for the main calculations for YOLO-V3 network. Unlike ResNet, which adds the values of the subsequent layers by constructing an identity map, DenseNet [53] connects all the layers for channel merging to achieve feature reuse. Compared with ResNet, the back propagation of the gradient is enhanced, which can make better use of feature information and improve the transmittance of the information between layers.The structure of DenseNet is shown in Figure 2.

In Figure 2, $x_{1}$, $x_{2}$, $x_{3}$, and $x_{4}$ represent the feature maps of the output layers, while $H_{1}$, $H_{2}$, $H_{3}$, and $H_{4}$ refers to the nonlinear transformations. The network contains l(l+1)/2 connections with l layers. Each layer is connected to all the other layers. Thus, each layer can receive all the feature maps of the preceding (l−1) layers. The feature map of each layer can be expressed in Equation (2).
(2)$$x_{l}=H_{l}[x_{0},x_{1},\ldots ,x_{l-1}]$$

The proposed densely connected network in this paper borrows from the idea of residual units in Figure 1. The convolution, Batch Normalization, and Leaky-ReLU make up the CBL module, while two CBL modules are cascaded into a Double-CBL (DCBL) module. We use the DCBL module as transport layer $H_{i}$: Conv ( 1 × 1 × 32)-BN-ReLU-Conv (3 × 3 × 64)-BN-ReLU and Conv (1 × 1 × 64)-BN- ReLU- Conv (3 × 3 × 128)-BN-ReLU. Thus, too many layers of DenseNet will lead the feature maps getting redundant and decrease the speed of detection, we set four layers for each module. The increment of the feature maps for each layer in module ’DENSE 1st’ is 64 while the increment of the feature maps for each layer in module ’DENSE 2nd’ is 128.

With the aim of reducing the network’s dependence on residual units, a part of the lower resolution layers of the feature extraction network is replaced by the improved densely connected network. The structure diagram of the proposed feature extraction network is shown in Figure 3.

To show the structure of our approach in detail, Table 2 gives the feature extraction network of our approach.

### 3.2. The Proposed Algorithm with Multi-Scale Detection

For an input image of 416 × 416, the size of the feature maps of the three detection layers are 13 × 13, 26 × 26, and 52 × 52, respectively. The smaller the size of the feature map is, the larger the area in the input image is in which each grid cell will correspond. On the contrary, the larger the size of the feature map is, the smaller the area in the input image is in which each grid cell will correspond. It means the 13 × 13 detection layer is suitable for detecting large targets, while the 52 × 52 detection layer is suitable for detecting small targets. Generally speaking, remote sensing images contain a large amount of small targets. In order to further enhance the detection performance of remote sensing targets, we need a larger-sized detection scale. The size of the new scale is 104 × 104. Compared with the original three scales, the four detecting scales strategy is suitable for detecting smaller-sized targets.

Furthermore, in order to avoid gradient fading, we replace the five convolutional layers with three residual units, which is in front of each detection layer. The structure of residual units and the proposed network are shown in Table 3 and Figure 4, respectively.

Table 1 and Table 2 show the structure of the feature extraction network of YOLO-V3 and our approach, respectively. Table 3 shows the structure of the residual units, which is in the end of four detection layers of our proposed network. In the end, Figure 4 exhibits the massive structure of our proposed network.

### 3.3. K-Means for Anchor Boxes

Inspired by Faster-RCNN, YOLO-V2 and YOLO-V3 introduced the ideal of the anchor box to predict the bounding boxes more accurately. In our approach, we ran K-means to generate the anchor boxes. The function of the K-means algorithm is conducting latitude clustering to make anchor boxes and adjacent ground truth have larger IOU values, which is not directly related to the size of anchor boxes.
(3)d(box,centroid)=1−IOU(box,centroid)

IOU refers to the intersection ratio and it is defined in Equation (4).
(4)$$\mathrm{IOU}=\;\frac{S_{overlap}}{S_{union}}$$

Soverlap refers to the overlap area between the predicted box and the ground truth and Sunion refers to the union area between them. The pseudocode of K-means in this paper is shown in Algorithm 1.
**Algorithm 1:** The pseudocode of K-means1: Given K cluster center points: (Wi,Hi),i∈{1,2,…,k},Wi,Hi refer to the width and height of each anchor box.2: Calculate the distance between each ground truth and each cluster center:d(box,centroid)=1−IOU(box,centroid). Since the position of the anchor box is not fixed, the center point of each ground truth is coincident with the clustering center.3: Recalculate the cluster center for each cluster: W′i=1Ni∑wi,H′i=1Ni∑hi4: Repeat step 2 and step 3 until the clusters converge.

We ran the K-means algorithm to get anchor boxes. In Figure 5, we can see the average IOU with a different number of clusters. The curve got more flat when the number increased. Since there are four detection layers in the network of our approach, we selected 12 clusters (anchor boxes). The sizes of the anchor boxes are as follows: (21, 24), (25, 31), (33, 41), (51, 54), (61, 88), (82, 91), (109, 114), (121, 153), (169, 173), (232, 214), (241, 203), (259, 271). Among them, (21, 24), (25, 31), (33, 41) are the anchor boxes for Scale 4. (51, 54), (61, 88), (82, 91) are the anchor boxes for Scale 3. (109, 114), (121, 153), (169, 173) are the anchor boxes for Scale 2 and (232, 214), (241, 203), (259, 271) are the anchor boxes for Scale 1.

### 3.4. Relative to the Grid Cell

When detecting the targets, we need to get the values of bounding boxes based on the predicted values. The process is shown in Figure 6. In Figure 6, $t_{x}$, $t_{y}$, $t_{w}$, and $t_{h}$ represent the predicted values of the network. $c_{x}$ and $c_{y}$ represent the offset of the gird relative to the upper left. The values of bounding boxes can be represented as:(5)$$\begin{array}{l} b_{x}=\sigma (t_{x})+c_{x}\\ b_{y}=\sigma (t_{y})+c_{y}\\ b_{w}=p_{w}e^{t_{w}}\\ b_{h}=p_{h}e^{t_{h}}\\ \sigma (x)=1/(1+e^{-x}) \end{array}$$

### 3.5. The NMS Algorithm for Merging Bounding Boxes 

Since there may be several bounding boxes corresponding to one target, the last step of our approach is to conduct non-maximum suppression (NMS) of the bounding boxes, which is aimed at eliminating unnecessary boxes. The steps of NMS are below.

Step 1: Take the bounding box with the highest confidence as the target for comparison. Then we compare the IOU between the bounding box and remaining boxes.Step 2: If the IOU is larger than the threshold we set, then remove the bounding box from the remaining bounding boxes.Step 3: Take the bounding box with the second highest confidence as the target for comparison and repeat Step 1 and Step 2 until all the bounding boxes are left.

The pseudocode of the algorithm is summarized in Algorithm 2.
**Algorithm 2:** The pseudocode of non-maximum suppression (NMS) for our approachOriginal Bounding Boxes:  $B=[b_{1},\ldots ,b_{M}]$, $C=[c_{1},\ldots ,c_{M}]$, threshold=0.6  B refers to the set of original bounding boxes  C refers to the set of confidences of B
Detection result:  F refers to the set of the final bounding boxes
1:  F←[]
2:   while B≠[] do:
3:     k←argmax C
4:     $F\leftarrow \;F.append(b_{k})\;$; $B\leftarrow \;del\;B\;[b_{k}]\;$; $C\leftarrow \;del\;C\;[c_{k}]\;$
5:     **for**
$b_{i}\in B$ do:
6:       **if**
$\mathrm{IOU}(b_{i},b_{k})\geq threshold$
7:        $B\leftarrow \;del\;B\;[b_{i}]\;$; $C\leftarrow \;del\;C\;[c_{i}]\;$
8:       **end**
9:     **end**
10:  **end**

## 4. Experiment and Results

In order to verify the validity of our improved YOLO-V3 for remote sensing target detection, we compared our approach with original YOLO-V3, YOLO-V3 tiny, and other state-of-the-art algorithms on RSOD and the USC-AOD dataset. The conditions of our experiment are as follows: Framework: python 3.6.5 and tensorflow 1.13.1, Operating system: Windows 10, CPU: i7-7700k, and GPU: NVIDIA GeForce RTX 2070. We set 50,000 training steps in this experiment. The learning rate of the model decreased from 0.001 to 0.0001 after 30,000 steps and to 0.00001 after 40,000 steps. We set the same parameters for other comparison algorithms. The initialization parameters of training lies in Table 4.

### 4.1. Loss Function

When training the network, loss function is used to measure the error between the predicted and true value. The loss function of the network can be defined in Equation (6).
(6)$$Loss=Error_{coord}+Error_{iou}+Error_{cls}$$

$Error_{coord}$ refers to a coordinate prediction error and it can be defined as:(7)$$\begin{array}{l} Error_{coord}=\lambda_{coord}\sum_{i=1}^{s^{2}}\sum_{j=1}^{B}I_{ij}^{obj}[{\left(x_{i}-{\bar{x}}_{i}\right)}^{2}+{\left(y_{i}-{\bar{y}}_{i}\right)}^{2}]\\ +\lambda_{coord}\sum_{i=1}^{s^{2}}\sum_{j=1}^{B}I_{ij}^{obj}[{\left(w_{i}-{\bar{w}}_{i}\right)}^{2}+{\left(h_{i}-{\bar{h}}_{i}\right)}^{2}] \end{array}$$

In Equation (7), $\lambda_{coord}$ refers to the weight of the coordinate error and we selected $\lambda_{coord}=5$ in our model. $S^{2}$ refers to the number of the grids (S×S). *B* refers to the number of bounding boxes per grid. ${Ι}_{ij}^{obj}$ refers to whether there is an object that falls in the *jth* bounding box of the *ith* grid cell. $\left({\bar{x}}_{i},{\bar{y}}_{i},{\bar{w}}_{i},{\bar{h}}_{i}\right)$ and $\left(x_{i},y_{i},w_{i},h_{i}\right)$ refer to the center coordinate, height, and width of the predicted box and the ground truth, respectively.

$Error_{iou}$ refers to an IOU error and it is defined as:(8)$$\begin{array}{l} Error_{iou}=\sum_{i=1}^{s^{2}}\sum_{j=1}^{B}I_{ij}^{obj}{\left(C_{i}-{\bar{C}}_{i}\right)}^{2}\\ +\lambda_{noobj}\sum_{i=1}^{s^{2}}\sum_{j=1}^{B}I_{ij}^{noobj}{\left(C_{i}-{\bar{C}}_{i}\right)}^{2} \end{array}$$

$\lambda_{noobj}$ refers to the confidence penalty when there is no object and we selected $\lambda_{noobj}=0.5$ in our model. $C_{i}$ And ${\bar{C}}_{i}$ refer to the true and predicted confidence, respectively.

$Error_{cls}$ refers to the classification error and it is defined as:(9)$$Error_{cls}=\sum_{i=1}^{s^{2}}\sum_{j=1}^{B}I_{ij}^{obj}{\sum_{c\in classes}\left(p_{i}(c)-{\hat{p}}_{i}(c)\right)}^{2}$$
where c refers to the number of classes of the targets.

### 4.2. The Evaluation Indicators

Based on the classification accuracy and prediction accuracy, the samples can be divided into four categories: TP (true positive), FP (fault positive), TN (true negative), and FN (fault negative). We define precision and recall in Equation (10) and Equation (11).
(10)$$\mathrm{Precision}=\frac{TP}{TP+FP}$$
(11)$$\mathrm{Recall}=\frac{TP}{TP+FN}$$

Mean average precision (mAP) is a performance metric for predicting target locations and categories. The accuracy and recall are mutually restricted in practice, and there will be ambiguity when compared separately. Therefore, in our experiment, we introduced mAP, which is one of the most important metrics to evaluate the performance of target detection algorithms.

### 4.3. Experiment on Remote Sensing Target Detection

The classifier trained based on a conventional dataset is not good at detecting remote sensing targets since remote sensing images have their particularities.

*Scale diversity*. Remote sensing images can be taken from hundreds of meters to nearly 10,000 meters in height, and ground targets may be of different sizes even if they are of the same kind. For example, ships in ports may be only tens of meters to more than 300 meters in size.*Perspective particularity*. The perspective of remote sensing images is basically overhead, but most of the conventional datasets are still ground level, so the mode of the same target is usually different. The detector trained well on the conventional datasets, which may have a poor effect on the remote sensing images.*Problem of small targets*. Most of the remote sensing targets are small in size. As a result, the target information is limited. The information of the targets has been lost due to the down sampling layers of the Convolutional Neural Network (CNN). After four times of down sampling, the feature map of the target with 24 × 24 pixels may take up only 1 pixel.*Problem of multi-directions*. The viewing angle of remote sensing images are usually overhead, while the directions of the targets are uncertain while there is a degree of certainty in conventional datasets.*The high complexity of the background*. The fields of remote sensing images are relatively large (usually covering several square kilometers). The fields of vision may contain various backgrounds, which will produce strong interference to the target detection.

Based on the above reasons, it is often difficult to train an ideal target detector from conventional datasets for target detection tasks of remote sensing images. A special remote sensing image database is needed.

#### 4.3.1. Dataset Analysis

Taking everything into consideration, we selected the RSOD and UCS-AOD dataset in the experiment. RSOD is the dataset of aerial images. It contains the targets of four categories: aircraft, playground, overpass, and oil tank. UCS-AOD is the dataset of target detection in aerial images. We generally consider the target, which the ground truth takes up less than 0.12% of the whole image as a small target. The ground truth takes up 0.12–0.5%, which is a medium target, and the ground truth takes up more than 0.5%, which is a large target. Of the four categories, the aircraft targets are mostly small in size. The oil tank targets are major of a small or medium size. The playground and overpass targets are big in size. The dataset includes targets under different lighting conditions and at different heights, and the shooting angles of the targets are also different.

Table 5 and Table 6 show the statistics of our remote sensing datasets. The targets in the dataset are mainly small or medium in size, and the distribution of the targets is relatively dense, which increases the difficulty of target detection. Figure 7 contains eight samples of the datasets in this paper. The targets in these samples are under a complex background. After a series of convolutional layers and down sampling layers, the targets take up even fewer pixels, which makes it difficult to detect them.

#### 4.3.2. Experimental Results and Analysis in RSOD and UCS-AOD Dataset

In order to compare the accuracy and real-time performance of the algorithms, the mAP and speed of our approach are evaluated. We compared our approach with the state-of-the-art target detection models in the RSOD dataset, and the comparison results are shown in Table 7. Furthermore, the comparison results of the targets with different sizes are shown in Table 8.

Table 7 shows that our approach is superior to other classical algorithms in the index of mAP. The detection speed is not significantly reduced relative to YOLO-V3. For aircrafts and oil tanks, which are mainly small and medium-sized targets, our approach has a clear improvement in detection accuracy compared to YOLO-V3. The experimental results show that our improved YOLO-V3 can effectively detect the remote sensing targets under the complex background in the condition of real-time detection. In Table 8, we divide target categories by size. We can see that our approach has more advantages than YOLO-V3 in detecting small-sized targets.

For the universality of our algorithm, we ran the experiment on the UCS-AOD dataset. The comparison results are shown in Table 9. In addition, from Table 8 and Table 9, we can see that the leak detection rate is significantly lower than YOLO-V3 and other state-of-the-art algorithms.

Under different backgrounds, partial detection results of our approach in RSOD and UCS-AOD dataset are shown in Figure 8. In the conditions of different illumination, different distributions, and different target sizes, our approach can detect the target accurately, which proves excellent detection performance for multi-scale remote sensing targets.

#### 4.3.3. Ablation Experiments

In this section, we need to verify the effectiveness of each improved module we proposed. In order to analyze the influence of module ‘DENSE 1st’ and module ‘DENSE 2nd’ (Figure 3) on the detection accuracy, different combination modes were set up in the experiment under the condition of three detection scales. The experimental results of each combination in the RSOD dataset are shown in Table 10.

It can be seen from the first experiment and the fourth experiment that the feature extraction network of the fourth experiment introduced dense connection modules based on Darknet53. mAP of its model improved from 77.10% to 84.38%. On the other hand, the detection speed of the fourth experiment increased from 29.7 FPS to 32.3 FPS compared to the first experiment. The experimental results show that our proposed feature extraction network can improve the performance of remote sensing target detection and also has advantages in detection speed.

In addition, Table 11 compared the experimental results of each module at the detection end based on an improved feature extraction network. It can be found in the comparison of the first experiment and the second experiment, and the comparison between the third experiment and the fourth experiment, that the fourth detection scale increased and improved mAP up to 5.95% and 5.78%, respectively. Among them, for the small-sized targets like aircraft, the accuracy is improved by 8.72% and 7.04%, respectively. This shows that the increased detection scale can effectively improve the detection accuracy of small targets. Compared with six convolutional layers, the ‘Res 3’ module can avoid gradient fading and reduce the number of parameters. The comparison of experiment 1 and experiment 3, and the comparison between experiment 2 and experiment 4 show that the ‘Res 3’ module can slightly increase the detection speed.

The ablation experiments result in Table 9 and Table 10, which proved that the improved feature extraction network and detection end we proposed can improve the feature extraction ability of the network and enhanced the detection accuracy of multi-scale remote sensing targets, especially small-sized targets. In addition, the detection speed of our approach is not significantly reduced when compared to YOLO-V3 and meets the real-time requirements.

#### 4.3.4. Expansion Experiment

In order to verify the effectiveness of our approach more intuitively, we selected several images and compared the detection results with YOLO-V3 and Faster RCNN. The comparison of the detection results are shown in Figure 9.

In Figure 9, a total of 24 detection results of eight groups were chosen in the RSOD dataset and UCS-AOD dataset to prove the superiority of the improved YOLO-V3. The pictures in the first list are the detection results of the YOLO-V3 network. The pictures in the second list are the detection results of Faster RCNN and the pictures in the third list are the detection results of our approach. It can be clearly seen that there are several small targets missed and detected by YOLO-V3. Although Faster RCNN performed better than YOLO-V3, leak detection still exists. On the other hand, all the targets were detected by our approach. The contrast experiments of eight groups and the ablation experiments showed that, by improving the feature extraction network and increasing the fourth detection scale, our approach enhanced the performance of detecting small targets with complex background conditions in remote sensing images.

## 5. Conclusions

In practical engineering applications, we need to consider both accuracy and speed of detection. The existing remote sensing target detection algorithms often fail to consider both of them. In this paper, we proposed an improved YOLO-V3-based model for multi-scale remote sensing target detection. On account of the complexity of the background of remote sensing targets, this puts forward a higher requirement for the ability of the network to extract features. In this paper, we focused on improving the original feature extraction network. Several improvements have been introduced to the original YOLO-V3 network. First, in order to extract feature information more effectively, a dense connection network (DenseNet) was introduced in the feature extraction network. Second, to enhance the performance of detecting small-sized targets, we extended the detection scales from 3 to 4. Third, we replaced three residual units with five convolutional layers, which is in each detection layer to avoid gradient fading. We can see from the ablation experiments that each improved module we proposed is effective in improving the detection accuracy. Experiments on RSOD and UCS-AOD datasets show that our approach achieves better performance on multi-scale remote sensing target detection. The improvement on the feature extraction network greatly improved the ability of extracting the features of the targets. The additional fourth detection scale strengthens the performance of detecting small targets. In the case of losing a portion of detection speed, the accuracy is greatly improved, especially for small remote sensing targets compared with YOLO-V3. Although numerous improved networks based on YOLO-V3 have been proposed, they usually detected targets in routine images. When facing complex remote sensing images, they did not do well. On the contrast, with the above measures adopted, our proposed algorithm is more suitable for remote sensing target detection than other state-of-the-art target detection algorithms. In further work, multi-receptive fields for the feature extraction of the network will be researched to boost the performance of remote sensing target detection. In addition, the latest version of YOLO: YOLO-V4 [55] has been proposed and this will be researched in further work.

## Figures and Tables

**Figure 1 sensors-20-04276-f001:**
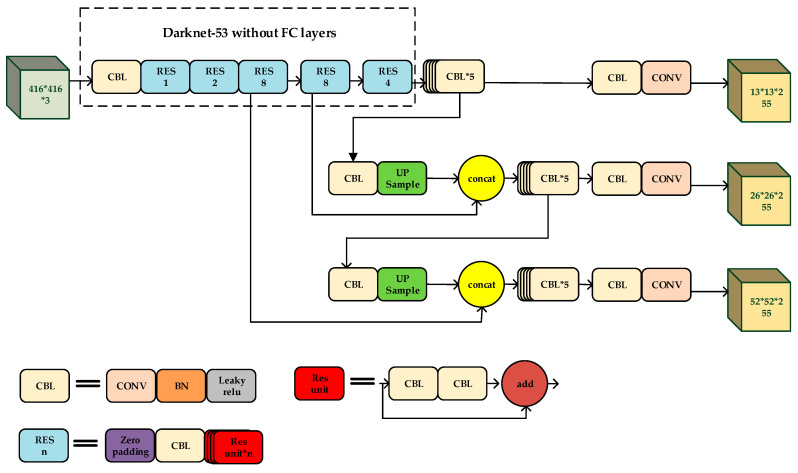
The network of You Only Look Once (YOLO)-V3.

**Figure 2 sensors-20-04276-f002:**
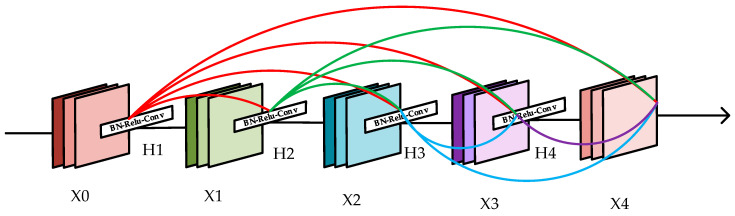
The structure of DenseNet.

**Figure 3 sensors-20-04276-f003:**
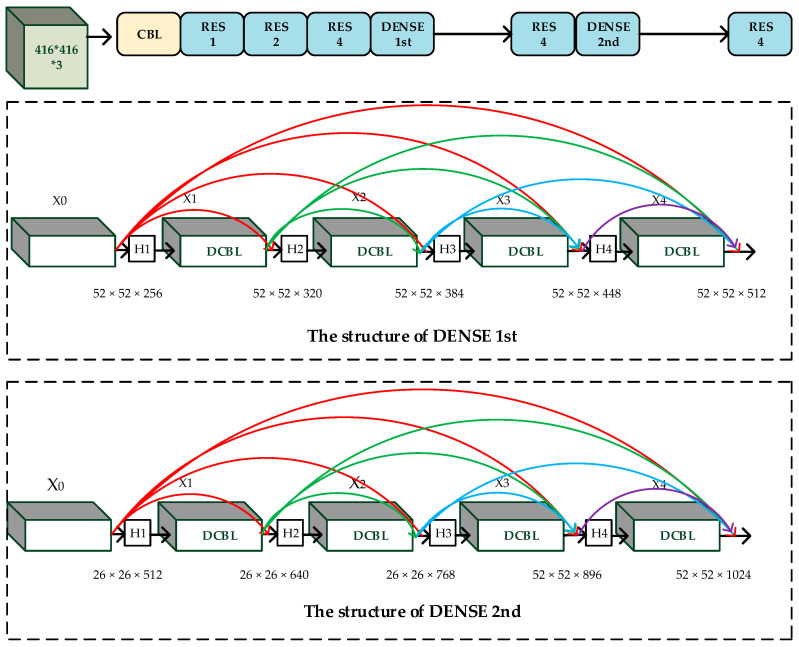
The structure diagram of the feature extraction network.

**Figure 4 sensors-20-04276-f004:**
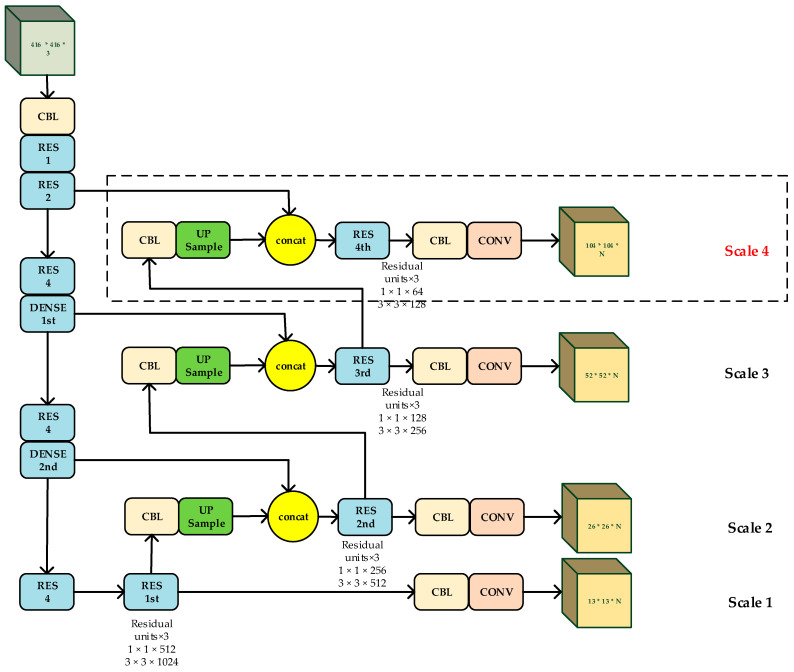
The Structure of the proposed network.

**Figure 5 sensors-20-04276-f005:**
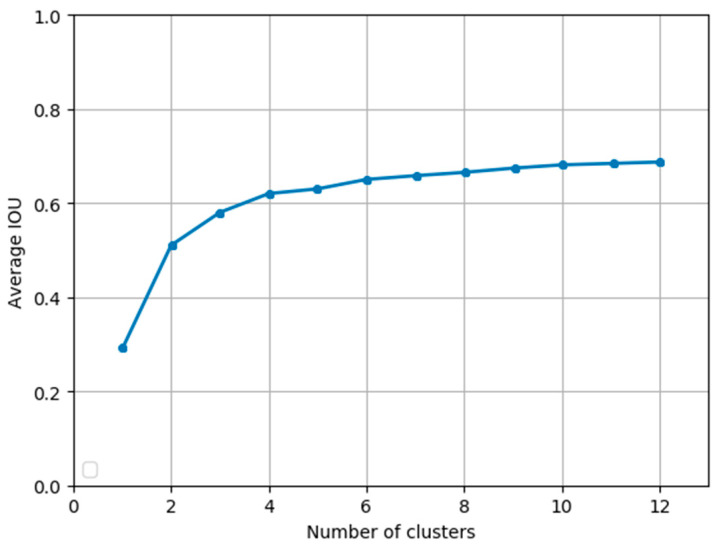
The relationship between the number of clusters and average IOU by K-means clustering.

**Figure 6 sensors-20-04276-f006:**
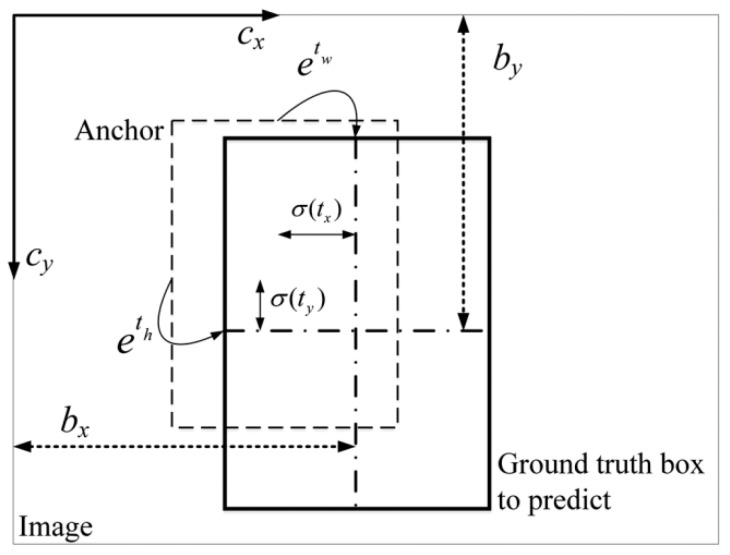
The final prediction.

**Figure 7 sensors-20-04276-f007:**
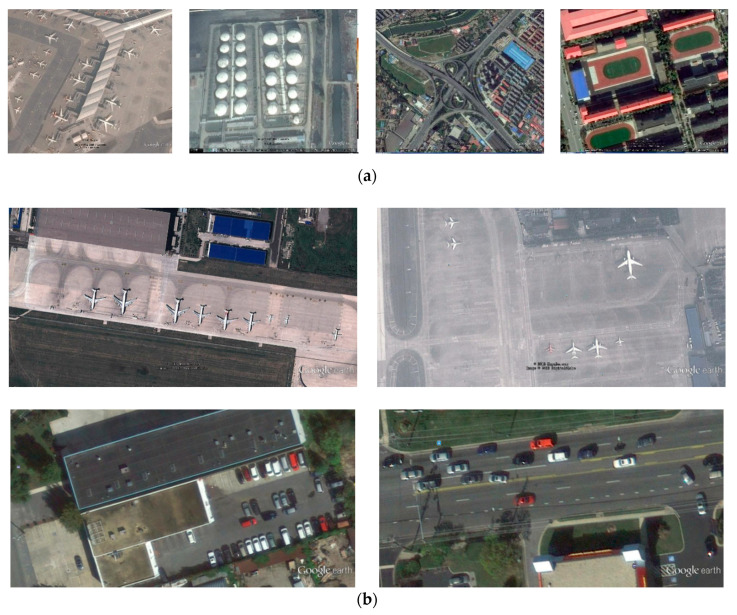
The samples of the datasets: (**a**) the samples of remote sensing object detection (RSOD) dataset and (**b**) the samples of dataset of object detection in aerial images **(**UCS-AOD) dataset.

**Figure 8 sensors-20-04276-f008:**
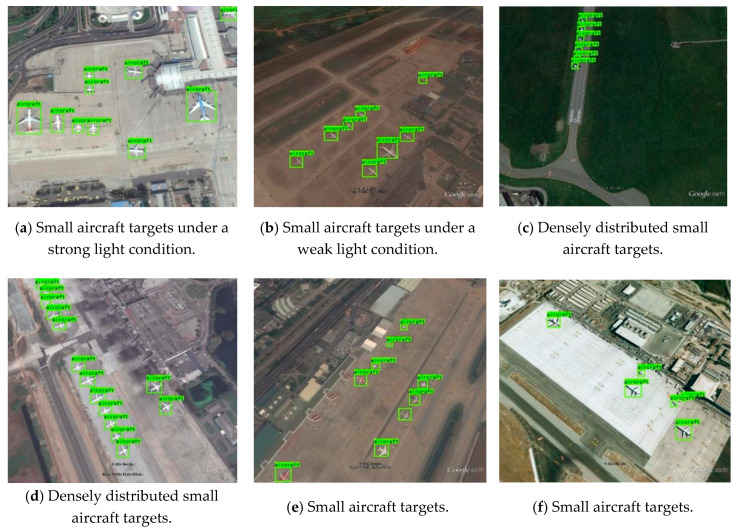
The detection results of the improved You Only Look Once (YOLO)-V3.

**Figure 9 sensors-20-04276-f009:**
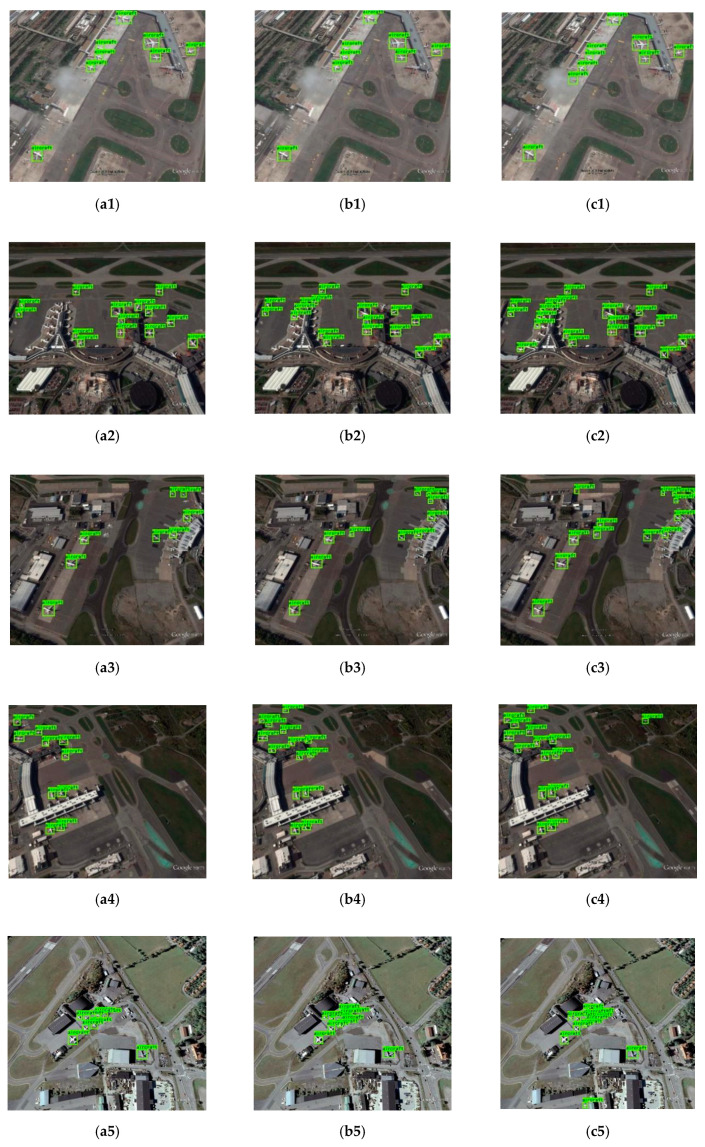
The comparison results of YOLO-V3 and our approach: (**a1**–**a8**) The detection results of YOLO-V3; (**b1**–**b8**) The detection results of Faster RCNN; (**c1**–**c8**) The detection results of our approach.

**Table 1 sensors-20-04276-t001:** The feature extraction network of You Only Look Once (YOLO)-V3.

	Layer	Filter	Size	Output
	Convolutional	32	3 × 3	416 × 416 × 32
	Convolutional	64	3 × 3/2	208 × 208 × 64
	Convolutional	32	1 × 1	
1×	Convolutional	64	3 × 3	
	Residual			208 × 208 × 64
	Convolutional	128	3 × 3/2	104 × 104 × 128
2×	Convolutional	64	1 × 1	
	Convolutional	128	3 × 3	
	Residual			104 × 104 × 128
	Convolutional	256	3 × 3/2	52 × 52 × 256
8×	Convolutional	128	1 × 1	
	Convolutional	256	3 × 3	
	Residual			52 × 52 × 256
	Convolutional	512	3 × 3/2	26 × 26 × 512
8×	Convolutional	256	1 × 1	
	Convolutional	512	3 × 3	
	Residual			26 × 26 × 512
	Convolutional	1024	3 × 3/2	13 × 13 × 1024
4×	Convolutional	512	1 × 1	
	Convolutional	1024	3 × 3	
	Residual			13 × 13 × 1024

**Table 2 sensors-20-04276-t002:** The feature extraction network of our approach.

	Layers	Filter	Size	Output
	Convolutional	32	3 × 3	416 × 416 × 32
	Convolutional	64	3 × 3/2	208 × 208 × 64
1×	Convolutional	32	1 × 1	
	Convolutional	64	3 × 3	
	Residual			208 × 208 × 64
	Convolutional	128	3 × 3/2	104 × 104 × 128
2×	Convolutional	64	1 × 1	
	Convolutional	128	3 × 3	
	Residual			104 × 104 × 128
	Convolutional	256	3 × 3/2	52 × 52 × 256
4×	Convolutional	128	1 × 1	
	Convolutional	256	3 × 3	
	Residual			52 × 52 × 256
4×	Convolutional	32	1 × 1	
	Convolutional	64	3 × 3	
	DenseNet			52 × 52 × 512
	Convolutional	512	3 × 3/2	26 × 26 × 512
4×	Convolutional	256	1 × 1	
	Convolutional	512	3 × 3	
	Residual			26 × 26 × 512
4×	Convolutional	64	1 × 1	
	Convolutional	128	3 × 3	
	DenseNet			26 × 26 × 1024
	Convolutional	1024	3 × 3/2	13 × 13 × 1024
4×	Convolutional	512	1 × 1	
	Convolutional	1024	3 × 3	
	Residual			13 × 13 × 1024

**Table 3 sensors-20-04276-t003:** The structure of residual units.

3×	Convolutional	512	1 × 1	
	Convolutional	1024	3 × 3	
	Residual (RES 1st)			13 × 13 × 1024
	The Structure of RES 1st			
3×	Convolutional	256	1 × 1	
	Convolutional	512	3 × 3	
	Residual (RES 2nd)			26 × 26 × 512
	The Structure of RES 2nd			
3×	Convolutional	128	1 × 1	
	Convolutional	256	3 × 3	
	Residual (RES 3rd)			52 × 52 × 256
	The Structure of RES 3rd			
3×	Convolutional	64	1 × 1	
	Convolutional	128	3 × 3	
	Residual (RES 4th)			104 × 104 × 128
	The Structure of RES 4th			

**Table 4 sensors-20-04276-t004:** The initialization parameters of training.

Input Size	Batch Size	Momentum	Learning Rate	Training Step
416 × 416	8	0.9	0.001–0.00001	50,000

**Table 5 sensors-20-04276-t005:** Remote sensing object detection (RSOD) dataset statistics.

Dataset	Class	Image	Instances	Target Amount		
				Small	Medium	Large
**Training** **Set**	Aircraft	446	4993	3714	833	446
	Oil tank	165	1586	724	713	149
	Overpass	176	180	0	0	180
	Playground	189	191	0	12	179
**Test** **Set**	Aircraft	176	1257	741	359	157
	Oil tank	63	567	257	213	97
	Overpass	36	41	0	0	41
	Playground	49	52	0	0	52

**Table 6 sensors-20-04276-t006:** Dataset of object detection in aerial images (UCS-AOD) dataset statistics.

Dataset	Class	Image	Instances
**Training Set**	Aircraft	600	3591
	Car	310	4475
**Test Set**	Aircraft	400	3891
	Car	200	2639

**Table 7 sensors-20-04276-t007:** Experimental comparison of accuracy and speed in the RSOD dataset.

Method	Backbone	Metric (%)					FPS
		Aircraft	Oil Tank	Overpass	Playground	mAP (IOU = 0.5)	
Faster RCNN	VGG-16	85.85	86.67	88.15	90.35	87.76	6.7
SSD	VGG-16	69.17	71.20	70.23	81.26	72.97	62.2
DSSD	ResNet-101	72.12	72.49	72.10	83.56	75.07	6.1
ESSD	VGG-16	73.08	72.94	73.61	84.27	75.98	37.3
YOLO-V2	DarkNet19	62.35	67.74	68.38	78.51	69.25	35.6
YOLO-V3	DarkNet53	74.30	73.85	75.08	85.16	77.10	29.7
YOLO-V3 tiny	DarkNet19	54.14	56.21	59.28	64.20	58.46	69.8
UAV-YOLO [52]	Figure 1 in [52]	74.68	74.20	76.32	85.96	77.79	30.1
DC-SPP-YOLO [54]	Figure 5 in [54]	73.16	73.52	74.82	84.82	76.58	33.5
ours	(Figure 3)	86.42	87.57	89.37	91.56	88.73	25.8

**Table 8 sensors-20-04276-t008:** Experimental comparison of accuracy measured by size.

Method	Backbone	Metric (%)			Leak Detection Rate (%)
		Small	Medium	Large	
Faster RCNN	VGG-16	84.73	87.87	89.18	11.8
SSD	VGG-16	70.38	73.41	77.51	21.1
DSSD	ResNet-101	74.42	75.18	77.70	15.2
ESSD	VGG-16	75.12	75.84	78.12	16.5
YOLO-V2	DarkNet19	63.20	68.53	69.28	24.3
YOLO-V3	DarkNet53	74.52	75.63	76.14	19.5
YOLO-V3 tiny	DarkNet19	55.26	56.47	60.17	31.4
UAV-YOLO [52]	Figure 1 in Reference [52]	75.45	75.15	76.85	17.1
DC-SPP-YOLO [54]	Figure 5 in Reference [54]	75.41	74.67	76.41	15.9
ours	(Figure 3)	87.51	87.93	90.23	10.2

**Table 9 sensors-20-04276-t009:** Experimental comparisons of accuracy and speed in the UCS-AOD dataset.

Method	Backbone	Metric (%)				FPS
		Aircraft	Car	Leak Detection Rate (%)	mAP (IOU = 0.5)	
Faster RCNN	VGG-16	87.31	86.48	13.8	86.90	6.1
SSD	VGG-16	70.24	72.61	23.7	71.43	61.5
DSSD	ResNet-101	73.17	74.19	16.1	73.68	5.2
ESSD	VGG-16	73.62	75.06	15.9	74.34	33.2
YOLO-V2	DarkNet19	63.17	68.42	23.0	65.80	34.3
YOLO-V3	DarkNet53	75.71	75.62	18.5	75.67	27.6
YOLO-V3 tiny	DarkNet19	57.58	56.35	35.2	56.97	65.3
UAV-YOLO [52]	Figure 1 in Reference [52]	75.12	75.60	16.5	75.36	28.4
DC-SPP-YOLO [54]	Figure 5 in Reference [54]	76.52	74.61	17.4	75.57	30.4
Ours	(Figure 3)	89.31	88.24	9.3	88.78	24.9

**Table 10 sensors-20-04276-t010:** Experimental comparisons of each combination in the feature extraction network.

	DENSE 1st	DENSE 2nd	Metric (%)					FPS
			Aircraft	Oil Tank	Overpass	Playground	mAP (IOU = 0.5)	
1			74.30	73.85	75.08	85.16	77.10	29.7
2	✓		76.81	75.38	77.21	85.37	78.69	30.9
3		✓	77.28	76.39	79.65	85.92	79.81	31.4
4	✓	✓	82.16	83.52	85.12	86.73	84.38	32.3

**Table 11 sensors-20-04276-t011:** Experimental comparisons of each combination in detection layers.

	4th Scale	Res 3	Metric (%)					FPS
			Aircraft	Oil Tank	Overpass	Playground	mAP (IOU = 0.5)	
1			77.25	76.38	84.36	86.12	81.03	29.7
2	✓		85.97	85.18	87.15	89.61	86.98	24.8
3		✓	79.38	78.85	85.29	88.28	82.95	30.1
4	✓	✓	86.42	87.57	89.37	91.56	88.73	25.8

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
