# Peer review of "Improved YOLO-V3 with DenseNet for Multi-Scale Remote Sensing Target Detection"

_sensors, 2020, doi:10.3390/s20154276_

Round 1
Reviewer 1 Report
Comments to the Authors
Target detection based on yolov3 in remote sensing images is an interesting and highly promising method for remote-target detection. This paper has some merits, but it also has marginal novelty and unclear expression of the work. My comments are listed as follows:
- The manuscript didn’t been added line number, so it is hard to review.
- In the introduction part, more existing methods of target detection of remote sensing images should be mentioned, and the introduction does not demonstrate the importance of this work.
- From Table 5, I find that accuracy of Faster-RCNN is similar to the proposed method, but the authors didn’t compare it in the Expansion Experiment. I would suggest adding the comparison results of Fast-RCNN with the proposed approach.
- The authors only test in RSOD dataset, I would suggest testing in another dataset such as UCAS-AOD.
- After a quick literature search, I found some other yolov3 approaches had been used in target detection of remote sensing images, please compare with them.
- Figure 4 looks like a screenshot version, please modify it.
- The authors mention the yolov3, I would suggest citing the original works, https://arxiv.org/abs/1804.02767.
- The manuscript seems to be a raw version and it has a lot of mistakes in format. For example, Page 2, Equal (1); Page 4, Second paragraph, and so on.
- The Font in figures should be consistent with the text, please modify it.
- There are some format mistakes in Table 4, Table 5, please modify it.
- The conclusion does not highlight the importance of the work.
Reviewer 2 Report
There are some comments for the authors to improve their paper which list as follows.
- It was better to explain the main contribution in the introduction, not in the abstract.
- Some acronyms were not defined clearly in the first time such as the RSOD dataset, mAP, ResNet, DenseNet.
- Please check all equation "*" should be "×".
- “So, to carry out feature fusion, the network uses up-sampling to keep the size of the
feature map down-sampled by 32× consistent with the feature map down-sampled by 16× and then merges the feature maps by concatenation. Similarly, we do the same for the feature map down-sampled by 16× and the feature map down-sampled by 8×.”Please make it a complete sentence, what is 32×…?
- Please always consistent in writing such YOLO-V2 or YOLO-v2? Check all the sentence.
- In the paper, the author says "In order to further enhance the detection performance of remote sensing targets, an extra scale is added to the network. The size of the new scale is 104×104. Compared with the original 3 scales, it is suitable for detecting smaller-sized targets." what is the reason of choosing the new scale 104x104?
- Please change the figure 4, it still has the red ruler lines.
- Please explain more about Figures, 4, 5, and 6 what is the relation between them?
- The sentence, font size and font style are messing. Please check it and write again carefully.
- Can you please explain more about the Experimental Results and Analysis, how you conduct the testing for your model? As all we know that, the detection result not always a success, sometimes also has miss detection. Do you use the same setting for each model (Faster RCNN, SSD, DSSD etc.…)?
- Please explain how you divide the size, small, medium and large? For example, small is <=32 px, large is >=96 px. The author says, “Of the four categories, the aircraft targets are smaller in size; the oil tank target is of mediumsize; and the playground and overpass targets are big in size.” In the other hand, it shows in Table 4 air craft and oil tank consist of small, medium, and large image. The data is confused. Please explain Table 4 in detail.
- As all we know that yolo is very bad at detecting small images. But the author says “The experimental results show that our improvedYOLO-V3 can effectively detect the remote sensing targets under the complex background in the condition of real-time detection and has more advantages than YOLO-V3 in detecting small targets.” Please explain about it. Also, in Table 4 it divides the target become small, medium, and large so when testing must base on the size as well and show in Table 5 and analyze the result. The experiment is not convincing enough, it is recommended to do more work.
- Please check all equation “Equal (1):,.. Equal (2)…Equal (10,11):...” I think it is must be equations.
- Please check all table and algorithm, for example:
The author writes Table 2. The detailed steps of NMS and Algorithm 2 NMS algorithm for our approach, please check it because it is not clear, if it is an algorithm just say algorithm but not a table.
- The author can also add a discussion about YOLO V4 in the manuscript.
- Some related references are missing, such as: [1]
[1]Hendry, Chen RC (2019) Automatic License Plate Recognition via sliding-window darknet-YOLO deep learning. Image and Vision Computing 87:47–56
Round 2
Reviewer 1 Report
I don't appreciate the response and work made by authors since the first version. There are still some things which should be polished.
- After a quick literature search, I found that there were many works for improving yolov3, the significance and advantages of this work should be highlighted in Conclusion and Introduction. Other works should be mentioned.
- Please check all the figures, there still are many mistakes in the figures.
- The descriptions of Table 12 and Table 13 should not be the same, please check it.
- In the expansion experiment section, the comparison of detection results is shown, and I suggest that authors select different datasets and different sizes of targets.
- Please check the format of the reference.
- Please check the grammar.
Reviewer 2 Report
I still have some minor check as follows:
1. Please check the Table 4 this is the Algorithm 1, not table please change. Also check Table 5 this is Algorithm 2.
2. Please be consistent in write YOLOV3 or YOLO-V3, check all.
3. Explain the abbreviation of YOLO in abstract.
Based on my personal opinion, after the author do the minor revision this paper can be considered for acceptance.
